


# Vulnerability analysis in Complex Networks under a Flood Risk Reduction point of view

Leonardo B. L. Santos[1,2], Aurelienne A. S. Jorge[3], Luciana R. Londe[1], Regina T. Reani[1], Roberta B. Bacelar[4], and Igor M. Sokolov[2]

[1]National Center for Monitoring and Early Warning of Natural Disasters (Cemaden), Sao Jose dos Campos/SP, Brazil
[2]Humboldt University of Berlin, Germany
[3]National Institute for Space Research (INPE), Cachoeira Paulista/SP, Brazil
[4]Anhanguera College, Sao Jose dos Campos/SP, Brazil

**Correspondence:** Leonardo B. L. Santos (santoslbl@gmail.com)

**Abstract.** The measurement and mapping of transportation network vulnerability constitute subjects of global interest. During a flood, some elements of a transportation network can be reached, causing damages directly (to people, vehicles and roads/streets) and indirect damages (services) with great economic impacts. The Complex Networks approach may offer a valuable perspective considering one type of vulnerability especially related to Disaster Risk Reduction on critical infrastructures: the topological vulnerability. The topological vulnerability index associated to an element in a graph is defined as the damage (variation) on the network's average efficiency due to the removal of that element. We have performed a topological vulnerability analysis to the highways in the state of Santa Catarina (Brazil), and produced a risk map considering that index and the flood susceptible areas. Our results can represent an important tool for stakeholders from the transportation sector.

## 1 Introduction

In a scenario of global change, extreme weather and climatic events are expected to increase in frequency and intensity and cause more social and economic impacts in several sectors, such as Critical Infrastructures, like transportation system and urban mobility. The assessment of the impacts of extreme weather conditions on transport systems shows high costs in several countries around the world (Pregnolato et al. (2017), Doll et al. (2014), Eidsvig et al. (2017)).

The Sendai Framework for Disaster Risk Reduction 2015-2030 is one of the most important documents in the Disaster Risk Reduction (DRR) guidelines. Among the seven global targets presented in this document, one refers to "Substantially reduce disaster damage to critical infrastructure and disruption of basic services".

For the Disaster Risk Reduction scientific community, vulnerability is a key concept. There are several types and meanings for vulnerability. According to Wisner (1994), vulnerability is "the characteristics of a person or group and their situation that influence their capacity to anticipate, cope with, resist and recover from the impact of a natural hazard (an extreme natural event or process)" (Wisner et al. (1994)).

For transportation literature, there are also different meanings for vulnerability (Schlogl et al. (2019)). Berdica (2002, p. 119) suggested that network vulnerability should be understood as "susceptibility to incidents that can result in considerable





reductions in road network serviceability" (Berdica (2002)). Taylor *et al.* (2006) understood network vulnerability as a concept close to network weakness and thus as the consequence of failure to provide sufficient capacity for the original purpose of the system (Taylor (2006)).

In this paper, we consider a topological meaning for vulnerability, under a Complex Network approach, as presented in (Yin & Xu (2010); Santos et al. (2019)). Interfaces between Complex Systems Science and Disaster Science were discussed in (Arosio et al. (2018)), however, the vulnerability index was not presented.

According to Pregnolato *et al.* (2016), network models are typically aspatial, the emphasis has been on topological interactions rather than considering the geography of the hazard and infrastructure (Pregnolato et al. (2016)).

Here, we use the concept of (geo)graph, a graph in a geographical space (Santos et al. (2017)), representing a set of highways as a graph, calculating the topological vulnerability of its elements and showing them on a map. We highlight the spacial location of the most vulnerable element, in order to combine those information with also the location of the most susceptible areas for flooding.

The development of a vulnerability mapping associated to floods and impacts on infrastructures is aligned with the 2030 Agenda for Sustainable Development, particularly with the Sustainable Development Goals (SDGs).

When natural hazards, such as flash floods or landslides, strike vulnerable areas, it is likely that local communities will struggle for coping with their effects. The lack of insurance, savings and loans, relief aid, and also inefficient government and slow decision making may reduce the potential for recovery for those communities (Chang & Falit-Baiamonte (2002)). Moreover, those affected communities with difficulties in the recovery process are likely to be more vulnerable to the next hazard, which intensifies economic and social problems ( (Fothergill & Peek, 2004), Cannon (1994), Carmo & Anazawa (2014)).

## 2 Material and Methods

### 2.1 Study area

Brazil is currently among the ten countries most affected by weather-related disasters in the last 20 years (UNISDR (2017)).

Santa Catarina state is located in the Brazilian South Region. There are 295 municipalities which are grouped in six regions: North, Itajaí valley, Florianópolis metropolitan area, South, Hills and West. The state population was 6.248.436 according to the last census track (2010). The State HDI - Human Development Index - is 0,774 and it is the third in the Brazilian HDI ranking (IBGE (2010)).

The Brazilian South Region - especially the states Santa Catarina and Rio Grande do Sul - is highly affected by disasters. Fash floods and floods correspond to 36,57% of these events in Brazil between 1991 and 2010 (CEPED UFSC (2012)). Indeed, Santa Catarina is the state with most decrees of emergency in its municipalities from 1991 to 2010 (CEPED UFSC (2012)).

Despite the high socio-economical indicators for municipalities from Santa Catarina, due to characteristics of occupation there are many communities at risk in those places (Londe et al. (2014)). The mountainous relief in the east side determined the human occupation in the fluvial plains, which are areas naturally prone to floods, and in the hills, which are areas prone to




landslides. Moreover, the industrialization and economic growth attracted more people to the regions and induced interventions in the environment, such as deforestation, landfill and non-regular constructions (SANTOS, 2012).

Herrmann *et al.* (2014) pointed that there were registrations of disasters in every month from 1980 to 2010 in Santa Catarina. He highlights 741 occurrences in January, 719 in May and 844 in July. There are high numbers of registers in summer and spring, associated to severe storms (Herrmann (2014)).

There is an annual mean of 64 registers of damages triggered by hydrological processes in Santa Catarina municipalities (Universidade Federal de Santa Catarina (2016)). The maximum value was achieved in 2008, when the material losses summed 4,9 billions of Reais (Universidade Federal de Santa Catarina (2016)).Between 1995 and 2014, the municipalities produced 2704 official documents to report damages and/or losses due to disasters in the State. The documents are often related to rainfall, especially flash floods (907 occurrences) (Universidade Federal de Santa Catarina (2016)).

## 2.2 Topological vulnerability

Critical Infrastructures, such as a network-type structure, can be represented (modeled) using a complex network approach. One complex network's measurement is particularly interesting in the context of critical infrastructures: the topological vulnerability (Santos et al. (2019)).

The shortest path length $d_{ij}$ between two nodes $i$ and $j$ is the smallest number of links from $i$ to $j$, among all the possible paths between $i$ and $j$.

The efficiency $e_{ij}$ in the communication between nodes $i$ and $j$ is inversely proportional to their shortest path length, i.e., $e_{ij} \sim 1/d_{ij}$. Let $E$ be the average efficiency of $G$, and let $V_k$ be the vulnerability associated with a node (or edge) $k$ of a graph $G$.

The vulnerability associated to element $k$ of graph $G$, $V_k$, is given by

$$V_k = \frac{E - E_k^\star}{E} \tag{1}$$

where $E_k^\star$ is the efficiency of the graph considering the removal of element $k$.

## 3 Results and Discussion

In Figure 1 is shown the vulnerability map for all highways in the study area.

As presented in Santos *et al.* (2019), the vulnerability associated to an element on a graph can be understood as the way a system reacts under a concrete threat on this element. Although it is a measurement associated to the element, assuming a possible value for each element, the vulnerability on complex networks brings information about the dynamics throughout the whole network.

In Figure 1 there are several element with low topological vulnerability, but there is some elements with vulnerability approx. 3% and an element with vulnerability approx. 5%, so, a flood enabling the traffic on this highway's element can reduce the

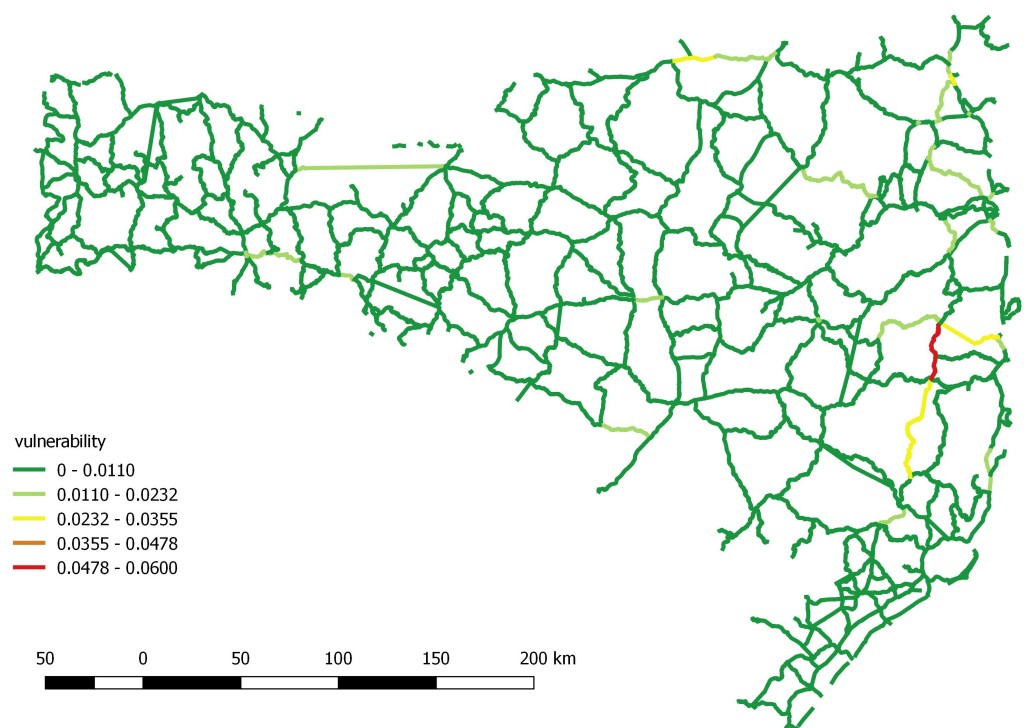

**Figure 1.** Vulnerability map for all highways in the study area. The green color is associated with the least vulnerable segments and the red color with the most vulnerable ones.

efficiency of this transportation network in 5%. It is important to highlight that in this region there is a heavy flow of people and goods, with some important national and international ports and airports.

In Figure 2 is shown the vulnerability map for a subset of the highways in the study area. Also, it is shown the areas most susceptible to floods and a digital image of the terrain. In this subset it is possible to see that there are some elements with

5  intermediate vulnerability close to urban flood areas.

To make a better urban planning and lessen the risk of disasters, the mapping of risk areas is an indispensable step. This mapping can be used to create a risk reduction plan, to define priority areas for attention in the municipalities, to make recommendations for works on infrastructure and to prepare municipal master plans.

The mapping of risk areas for Santa Catarina comprises:

10    – Elaboration of preliminary maps of susceptibility to hydrological events


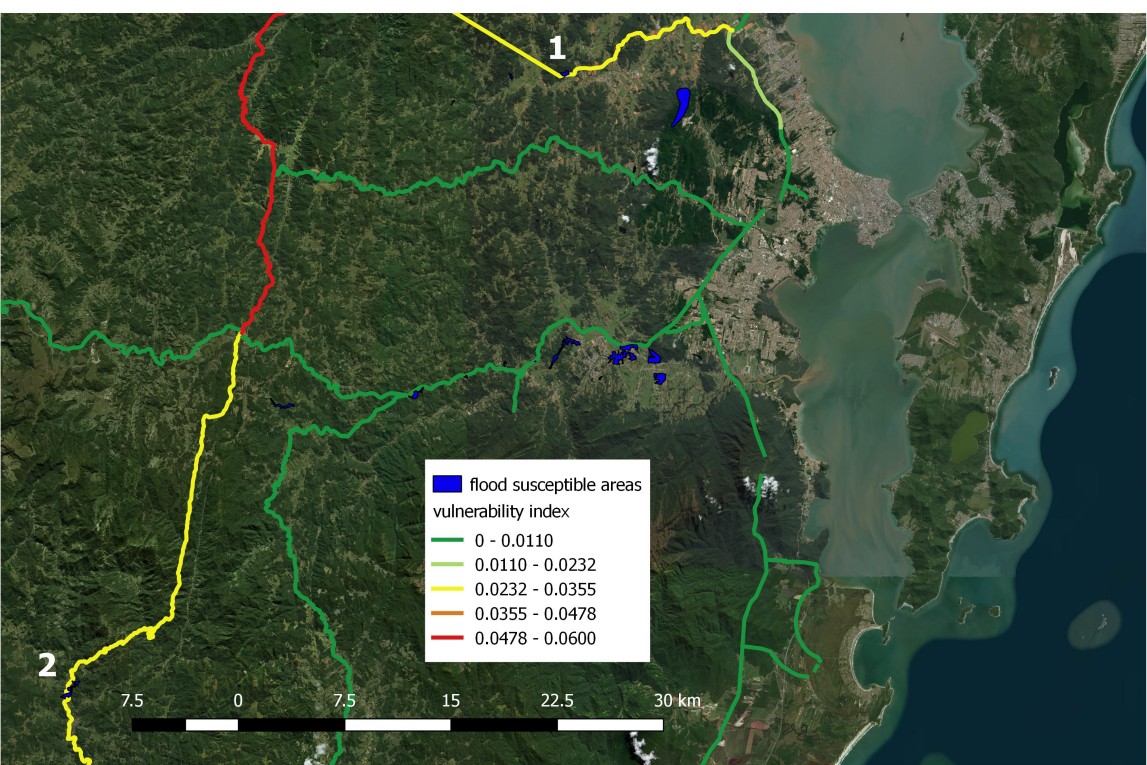

**Figure 2.** TEXT

- Grouping by sectors of high and very high risk for each municipality, with detailed maps for areas where there is already a risk

- Mapping of hazards, according to methodological aspects from GIDES Project (GIDES (2014))

- Elaboration of final maps of susceptibility to hydrological events for the monitored municipalities.

5    Santa Catarina follows the guidelines established from GIDES Project, which is a partnership between Brazil and Japan to strengthen the National Strategy for integrated Management of Risks and Disasters. The project's goal is to reduce risks of disasters through non-structural preventive actions. The mains results are the improvement of assessment systems and risk mapping, warnings and urban planing for disaster prevention.



## 4   Conclusions

From the Disaster Risk Reduction point of view, the vulnerability index can be a good proxy of potential impacts, by showing the elements that, if removed, could cause the biggest impacts on the efficiency of the network.

  In this paper, we modeled the set of highways from our study area as a graph and calculated some topological index, as the
topological vulnerability one. Using the (geo)graph approach (Santos et al. (2017)), it was possible to represent the results in a Geographical Information System.

  It was possible to see that there are some elements with vulnerability of approx. 5%, so, a flood enabling the traffic on this highway's element can reduce the efficiency of this transportation network in 5%. Also, there are some elements with intermediate vulnerability close to urban flood areas. In the study area, the State of Santa Catarina, in Brazil, there is a heavy
flow of people and goods, with some important national and international ports and airports.

  In the broader context, this work could support the "inclusive and sustainable development" pillar of the 2016 Country Partnership Framework established by the World Bank for Brazil by identifying areas of deprivation and making vulnerability to disasters an important consideration in the targeting of socioeconomic development.

  A possible extension for this investigation is to draft risk scenarios considering other components, such as the dynamic
exposure (daily traffic on each highway) and other kinds of vulnerability, for example, one based on the road pavement.

*Author contributions.*   These authors contributed equally to this work.

*Competing interests.*   No competing interests are present

*Acknowledgements.*   Funding: São Paulo Research Foundation (FAPESP), Grant Number 2015/50122-0 and DFG-IRTG Grant Number 1740/2; FAPESP Grant Number 2018/06205-7; CNPq Grant Number 420338/2018-7



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
