# Peer review of "Vulnerability analysis in Complex Networks under a Flood Risk Reduction point of view"

_Natural Hazards and Earth System Sciences, 2019_

## Referee Comment (RC1) · Anonymous Referee #1 · 21 Jul 2019

General comment:

In general, the work is too abstract, non-informative, and thus it is impossible to regenerate and validate the results. The proposed methodology is not innovative either as there are many researches devoted to the application of graph theory to vulnerability assessment of network infrastructures (transportation, power grids, water distribution pipeline, tank terminals, etc.).

Major comments:

1. The authors seem to have mistaken the criticality (importance) metric for the vulnerability metric. For instance, the relationship presented in Equation (1) is so much analogous to the Fussell-Vesely importance measure used in cut set analysis, or the

definition given in line 25 is close to criticality of an element rather than its vulnerability. I think the authors need precisely define what they mean by vulnerability before they apply it to their case study. 2. Despite its heading, Section 2.1 does not provide any information about the transport network of the area under study or the severity and characteristics of potential floods. 3. I am not sure about the accuracy of Fig. 1 because, (i) only one segment is identified as the most vulnerable segment (denoted with color red) in such a complicated network of roads, and (ii) there is no second most vulnerable segment (denoted with color orange). 4. The conclusions are not well supported by the study. For instance, how have the authors concluded, without any sensitivity analysis or validation, that their proposed vulnerability metric is a good proxy of the potential impacts of floods?

Minor comments:

5. The manuscript's English need proofreading; there are many typos, grammatical errors, etc. 6. The manuscript's title is inaccurate since the work is only focused on transportation network not all other types of complex networks. 7. Caption of Fig. 2 is missing. 8. Line 29-30: what is meant by "vulnerability approx."? 9. Line 18: since the work is about critical infrastructures, the definition of "people/group vulnerability" is rather irrelevant.

---

## Referee Comment (RC2) · Anonymous Referee #2 · 9 Sep 2019

Paper Review Feedback General Comments: The manuscript considers complex networks approach for analysing topological vulnerability of a transportation network at a case study of the state of Santa Catarina (Brazil) within a context of reducing flood risk. While the conceptualisation of the research problem may be of interest to the journal's readers, there is a lack of sufficient detail and clarity on the method applied (e.g., how vulnerability is assessed) and indication of the validity of the results/conclusions. There are also major grammar related issues throughout the paper and requires a thorough proofreading. A summary of some specific comments and example technical corrections are outlined below for the authors to consider. Specific Comments: ïĆğ Abstract: o 1st sentence: it would be good to specify in terms of 'vulnerability' to what? o 2nd sentence: Do the authors mean '. . .some elements of a transportation network cannot

be reached, . . .'? o 4th sentence: what does 'in a graph' refer here? Is it necessary to have it? Otherwise, need clarification. o Last sentence: '. . .an important tool for stakeholders. . .?' It would be good to clarify 'tool for what purpose'? ïĆğ Introduction: o This section is too broad and needs to focus on providing useful context to the specific topics explored – transportation network and implication of (vulnerability to) flooding. For example, there is a passing comment on '. . .lack of insurance, savings and loans, . . .', and it is not clear how this is relevant to the focus of the paper. ïĆğ Methods, and Results and Discussion: o This section lacks sufficient detail (except directing to another paper) to clearly understand the methodology employed in the vulnerability analysis. For example, it's not exactly clear how 'efficiency' (both at 'full capacity' and after 'removal of an element') is measured/estimated. Is it just an inverse of the length of an element? If so, why is this considered as a measure of 'vulnerability'? Also, not clear vulnerability to what? How the result shown in Figure 1 is produced also need to be explained in detail. o It is also not clear how the 'flood susceptible areas' (shown in Figure 2) are identified/modelled – the method used needs to be clearly stated? In addition, the paper's title suggests the vulnerability analysis is conducted within a viewpoint of flood risk reduction, but from the result in Figure 2 it is not clear what exactly the link is between 'vulnerability' (calculated based on just 'efficiency' as stated above) and 'flooding'. For example, should the vulnerability analysis consider identifying road network elements that are affected by (vulnerable to) flooding? That is, the index used to measure vulnerability needs to include a measure of the flood risk, than just efficiency. If this is the approach used, authors need to clarify this. As it stands, the link appears to be based on 'proximity analysis', if this is the case, the paper also need to clarify this. ïĆğ Conclusions: o This section should be less about just a repeat summary of the results from the previous sections, but more on a critique of the method and setting the results in the context of existing literature. This is important to highlight the validity of the method applied and also similarities and differences in conclusions of this study with other studies. Technical corrections: ïĆğ Introduction: o P2, Line 1–3: Check citation consistency/format: Taylor et al. (2006) vs. Taylor (2006)? o P2, Line 10–12:

Statement needs clarification – element vulnerability to what and how is it different from 'most susceptible areas for flooding'? o P2, Line 15–16: General statements like this should either be minimised/avoided or need to be supported with evidence (e.g., reference). ïĆğ Methods: o P3, Line 3–5: Check citation consistency/format: Herrmann et al. (2014) vs. Herrmann (2014)??? ïĆğ Results and Discussion: o P4, Figure 1: consider better/representative vulnerability classes (than which appears to be based on 'natural breaks'?). o P5, Figure 2: Same as above comment. Also, there is no caption. ïĆğ References: o Check citation formats in main text – e.g., with double brackets in a number of places. o Consult appropriate referencing style: the order of the list needs to be either in order of appearance in the main text (in which case, it should be numbered) or alphabetically – at the moment, it's neither.
* * *

---

## Referee Comment (RC3) · Anonymous Referee #3 · 9 Sep 2019

The submitted manuscript presents the concept of Geo-graph and use it to model vulnerability for route access. Some comments and questions are presented below.

Comments: * The abstract should include highlighting results beyond mention "Our results can represent an important tool for stakeholders from the transportation sector."; * Page 1, Line 20: double citation for a same author; * Second and third paragraphs in introduction section may be collapsed. The authors are invited to check similar occurrences on next paragraphs; * Please, check the journal directives about citing/referencing – as written on Page 2/Line 4, "...as presented in (Yin & Xu (2010); Santos et al. (2019))", the included parenthesis looks inadequate for this kind of citation. The authors should consider this concern for the entire manuscript; * Furthermore, paragraph of single phrase should be avoided (e.g., Page 2, Line 7 and more); * Last

paragraph of Section 1 looks out of context. The authors may rise such problematic before state the use of Geo-graphs as a tool on flooding situations; * Any previous study or similar research (on the Geo-graph point of view) were cited. If this is a pioneer study, the authors should ensure and then highlight it; * Page 2, Line 25: check decimal/thousands separator on English writing; * Include a figure to express the study area location (South america → Brazil → State) may help the study/manuscript understanding; * Page 3, Line 3: Since Herrmann et al. (2014) has more then one author, the pronoun "He" is inadequate; * Page 3, Line10: Use acronym for institution citation, instead of Universidade Federal de Santa Catarina; * Regarding a path dij, always is possible to find j from i? Are defined paths with null cost? If yes, how the efficiency is computed in such cases? Why the efficiency is inversely proportional to dij? The proportional symbol was wrongly chosen; * A discussion about the vulnerability on k (the meaning behind the mathematical definition) should be included – what such model means? * Page 3, Line 30: What means a vulnerability of 3%? How such value is interpreted/understood? * How Figure 1 was generated? The author may include a graph representation for "lengths" information used to achieve Figure 1; * Discussions regarding Figure 2 should be improved/enhanced; * How the flood susceptible areas on Figure 2 were obtained? How such areas affect the path's susceptibilities? Low susceptibilities are found near to flood areas – isn't expected the inverse behavior? * Figure 2 caption is missing. What means the numbers 1 and 2 on Figure 2? * The last paragraph of Section 3 looks out of context; * The conclusions should be improved/enhanced;

---

## Author Comment (AC1) · 15 Oct 2019

Dear Editor and dear Reviewers,

Dear Editor, many thanks for having forwarded us the comments of your Journal's reviewers on our Manuscript "Vulnerability analysis in Complex Networks under a Flood Risk Reduction point of view".

Our colleagues have reviewed our manuscript and evaluated our results positively, as seen from their reviews. Moreover, they have offered us a series of precious suggestions and comments on how to improve the quality and clarity of our work, suggestions and comments that we have gladly used in the revision process of the Manuscript.

Here, we enclose all relevant modifications to be done in our revised version of the

[Figure]

Manuscript, together with our answers to the Referees (in italic).

We thank you in advance for the attention you will devote to our resubmission.

Kind regards, The Authors

**REVIEWER1**

The manuscript's title is inaccurate since the work is only focused on transportation network not all other types of complex networks.

R: We thank the reviewer for this comment. We consider this new title "Vulnerability analysis in Transportation Complex Networks under a Flood Risk Reduction point of view" to be proposed to the Editor.

The proposed methodology is not innovative either as there are many researches devoted to the application of graph theory to vulnerability assessment of network infrastructures (transportation, power grids, water distribution pipeline, tank terminals, etc.).

R: Absolutely, "there are many researches devoted to the application of graph theory to vulnerability assessment of network infrastructures (transportation, power grids, water distribution pipeline, tank terminals, etc.)", however, as far as we know, there is no one published paper using the complex network measure "vulnerability" in transportation networks under a Disaster Risk Reduction point of view (including, for example, susceptibility maps) - and this is an innovative approach presented in this paper.

the relationship presented in Equation (1) is so much analogous to the Fussell-Vesely importance measure used in cut set analysis, or the definition given in line 25 is close to criticality of an element rather than its vulnerability

R: The pointwise vulnerability index in complex network science was defined in Goldshtein et al. (2004), citing two relevant previous works: Latora & Marchiori (2001) and Latora & Marchiori (2004). In a new version, we are going to clarify it.

This vulnerability measure is not directly analogous to the Fussel-Vesely importance

measure nether really close to criticality of an element. We thank the reviewer for this comment, and will consider as perspective to prepare a review paper about these conceptual relations among different measures.

Section 2.1 does not provide any information about the transport network of the area under study or the severity and characteristics of potential floods.

R: Some paragraphs about it will be included in the manuscript.

"There are 295 municipalities and more than 6 millions inhabitants in the State, according to the last census track (2010). The State HDI - Human Development Index - is 0.774 and it is the third in the Brazilian HDI ranking (IBGE, 2010). Despite the high socio-economical indicators for municipalities from Santa Catarina, due to characteristics of occupation there are many communities at risk in those places (Londe et al., 2014). The mountainous relief in the east side determined the human settlement in the fluvial plains, which are areas naturally prone to floods. Moreover, industrialization and economic growth attracted more people to the regions and induced interventions in the environment, such as deforestation, landfill and non-regular constructions (Londe et al., 2014)."

"Among the highways in Santa Catarina State, the BR-280 is one of the most important one, playing an important role in the flow of products to the ports of São Francisco do Sul, Itajaí and Paranaguá, as well as promoting the interconnection between important cities in the region, such as Joinville and Jaraguá do Sul. Thus, there is a large flow of people and goods on this highway. The susceptible flood areas used in this study were mapped by the Geological Survey of Brazil (CPRM), based on a database of previous occurrences and in situ evaluation of physical characteristics (CPRM, 2019)."

I am not sure about the accuracy of Fig. 1 because, (i) only one segment is identified as the most vulnerable segment (denoted with color red) in such a complicated network of roads, and (ii) there is no second most vulnerable segment (denoted with color orange).

R: The color scale adopted was regular over the range of vulnerability values: each color's "interval length" was equal to the others. In a new version, we are going to prepare a "quantile-based" scale.

The conclusions are not well supported by the study. For instance, how have the authors concluded, without any sensitivity analysis or validation, that their proposed vulnerability metric is a good proxy of the potential impacts of floods?

R: This paper aims to discuss some data based on the scientific literature on disaster concepts. Vulnerability itself does not create the impact: a disaster is triggered by a hazard, such as heavy rain. For data in this paper, the rain (triggering factor) should be stronger than that predicted in the engineering highway project. Nevertheless, vulnerability is a reasonable conceptual proxy for disaster impacts.

Also, it is important to highlight that this paper is about a "topological vulnerability", and there are, of course, several "other vulnerabilities", as social/economic and engineering-related ones. In a new version, we are going to include a new paragraph about it in the Conclusions Section.

Line 18: since the work is about critical infrastructures, the definition of "people/group vulnerability" is rather irrelevant R: This paper is also about disasters (floods) and it is not possible to discuss disasters without some considerations about vulnerability, once vulnerability is one of the main elements in risk analysis. The authors consider that transportation networks are an infrastructure for a social process: mobility. For a disaster risk reduction holistic analysis, it is important to consider different kinds of vulnerability. In this paper, the topological vulnerability was considered. Other kinds of vulnerability will be incorporated in future investigations, as an engineering vulnerability and a people/group/social vulnerability. This point will be considered in the perspective section in a reviewed version.

Caption of Fig. 2 is missing

R: In a new version, the caption will be included.

"Vulnerability index map for a subset of highways in the study area. The red color with the most vulnerable segments. The blue polygons are flood susceptible areas".

Line 29-30: what is meant by "vulnerability approx."?

R: In a new version, this abbreviation will be removed.

We do thank the reviewer for all the comments, the contributions were very important to improve this work.
* * *
* * *
**REVIEWER2**

Paper Review Feedback General Comments: The manuscript considers complex networks approach for analysing topological vulnerability of a transportation network at a case study of the state of Santa Catarina (Brazil) within a context of reducing flood risk. While the conceptualisation of the research problem may be of interest to the journal's readers, there is a lack of sufficient detail and clarity on the method applied (e.g., how vulnerability is assessed) and indication of the validity of the results/conclusions.

R: In a new version, these points will be clarified, based on these replies to the reviewers.

There are also major grammar related issues throughout the paper and requires a thorough proofreading.

R: In a new version, this problem will be solved.

A summary of some specific comments and example technical corrections are outlined below for the authors to consider. Specific Comments: Abstract: o 1st sentence: it would be good to specify in terms of 'vulnerability' to what?

R: to natural hazards.

o 2nd sentence: Do the authors mean ': : :some elements of a transportation network cannot be reached, : : :'?

R: In a new version, we are going to exchange the word "reached" by "affected".

o 4th sentence: what does 'in a graph' refer here? Is it necessary to have it? Otherwise, need clarification.

R: In a new version, these terms will be withdrawn.

o Last sentence: ': : :an important tool for stakeholders: : :?' It would be good to clarify 'tool for what purpose'?

R: In a new version, we are going to clarify it:

"The Risk Knowledge, combining hazard and vulnerability, is the first pillar of an Early Warning System. Risk maps can represent an important tool for stakeholders from the transportation sector in a climate change, disaster risk reduction and sustainable development agenda."

Introduction: o This section is too broad and needs to focus on providing useful context to the specific topics explored – transportation network and implication of (vulnerability to) flooding. For example, there is a passing comment on ': : :lack of insurance, savings and loans, : : :', and it is not clear how this is relevant to the focus of the paper.

R: We thank the reviewer for this comment. In a new version, we are going to clarify it, based on a literature overview:

"Among several works about vulnerability in transportation networks, Yin & Xu (2010) showed a case study using the vulnerability index in real transportation networks, but not under a disaster science context. Recently, Mattsson & Jenelius (2015), Arosio et al. (2018) and Santos et al. (2019b) discussed interfaces between Complex Systems Science and Disaster Science. However, Mattsson & Jenelius (2015) did not apply

its ideas in any real case study, Arosio et al. (2018) did not analyse the topological vulnerability index, and Santos et al. (2019) did not show any susceptibility map, just the topological vulnerability index itself."

"To fill these gaps, this paper presents a case study applying the vulnerability index, as used in the complex network science, in transportation networks under a Disaster Risk Reduction point of view, drawing risk scenarios considering both the generated vulnerability maps and also susceptibility maps. Our main scientific questions are: where are the most vulnerable links in a transportation network? Are they close to the most susceptible flood areas?"

Methods, and Results and Discussion: o This section lacks sufficient detail (except directing to another paper) to clearly understand the methodology employed in the vulnerability analysis. For example, it's not exactly clear how 'efficiency' (both at 'full capacity' and after 'removal of an element') is measured/estimated. Is it just an inverse of the length of an element? If so, why is this considered as a measure of 'vulnerability'? Also, not clear vulnerability to what?

R: It is important to consider the huge challenge of multi/inter-disciplinary studies, which is making different knowledge areas "talk the same language" (or to talk in a way it is possible to understand each other). One of the aims in this manuscript is to adjust these languages, in a way we can combine Complex Network Science and Disaster Science - this one including many definitions and concepts from Social Sciences.

The efficiency associated with a link between two nodes is, in the Complex Network Science point of view, represented as the inverse of the path's length between these two nodes. The vulnerability, in the Complex Network Science point of view, is an index associated to an element, that brings us information about the impact (on the average network's efficiency) of the removal of this element.

Combining Disaster Science and Network Science, it is possible to say that the vulnerability index, in Network Science, can be used to estimate the vulnerability, in Disaster

[Figure]

Science, associated to an element considering its removal. For example, when a highway can not be used after a flood episode, the vulnerability complex network index addresses the impact of it (this "removed" highway) in the average efficiency of the highway system (as a set/network of highways). It's important to highlight that it is a "topological vulnerability", and there are, of course, several "other vulnerabilities", as social/economic and engineering-related ones.

How the result shown in Figure 1 is produced also need to be explained in detail.

R: Using the (geo)graph approach, we have represented the set of highways as a graph, calculated the properties and represented them back on the highways map. We are going to clarify it in the manuscript.

It is also not clear how the 'flood susceptible areas' (shown in Figure 2) are identified/modelled – the method used needs to be clearly stated?

R: The susceptible flood areas used in this study were mapped by the Geological Survey of Brazil (CPRM), based on a database of previous occurrences and in situ evaluation of physical characteristics.

In addition, the paper's title suggests the vulnerability analysis is conducted within a viewpoint of flood risk reduction, but from the result in Figure 2 it is not clear what exactly the link is between 'vulnerability' (calculated based on just 'efficiency' as stated above) and 'flooding'. For example, should the vulnerability analysis consider identifying road network elements that are affected by (vulnerable to) flooding? That is, the index used to measure vulnerability needs to include a measure of the flood risk, than just efficiency. If this is the approach used, authors need to clarify this. As it stands, the link appears to be based on 'proximity analysis', if this is the case, the paper also need to clarify this

R: The Risk Knowledge, combining hazard and vulnerability, is the first pillar of an Early Warning System. Risk maps can represent an important tool for stakeholders from

the transportation sector in a climate change, disaster risk reduction and sustainable development agenda.

Conclusions: o This section should be less about just a repeat summary of the results from the previous sections, but more on a critique of the method and setting the results in the context of existing literature. This is important to highlight the validity of the method applied and also similarities and differences in conclusions of this study with other studies.

R: In a new version, we are going to include some new paragraphs about it.

"There are some elements with vulnerability index of approximately 5%, therefore a flood enabling the traffic on this highway's element can reduce the efficiency of this transportation network by approximately 5%. Also, there are some elements with high vulnerability index close to urban flood areas, for example, in the cities of Mafra and Rio Negrinho, where the BR-280 highway crosses the Negrinho River. This area is marked by several records of floods in the rainy season (hazard component), which makes traffic in the region unfeasible (impact). In the study area, the State of Santa Catarina, in Brazil, there is a heavy flow of people and goods, with some important national and international ports and airports."

Technical corrections: ï′C ËŸg Introduction: o P2, Line 1–3: Check citation consistency/format: Taylor et al. (2006) vs. Taylor (2006)?

P2, Line 15–16: General statements like this should either be minimised/avoided or need to be supported with evidence (e.g., reference).

Methods: o P3, Line 3–5: Check citation consistency/format: Herrmann et al. (2014) vs. Herrmann (2014)???

P2, Line 10–12: Statement needs clarification – element vulnerability to what and how is it different from 'most susceptible areas for flooding'?

P5, Figure 2: Same as above comment. Also, there is no caption.

References: o Check citation formats in main text – e.g., with double brackets in a number of places.

Consult appropriate referencing style: the order of the list needs to be either in order of appearance in the main text (in which case, it should be numbered) or alphabetically – at the moment, it's neither.

R: In a new version, we are going to correct all these technical mistakes.

Results and Discussion: o P4, Figure 1: consider better/representative vulnerability classes (than which appears to be based on 'natural breaks'?)

R: The color scale adopted was regular over the range of vulnerability values: each color's "interval length" was equal to the others. In a new version, we are going to prepare a "quantile-based" scale.

We do thank the reviewer for all the comments, the contributions were very important to improve this work.
* * *
**REVIEWER3**

The submitted manuscript presents the concept of Geo-graph and use it to model vulnerability for route access. Some comments and questions are presented below.

Comments: *The abstract should include highlighting results beyond mention "Our results can represent an important tool for stakeholders from the transportation sector."

R: Absolutely:

"The Risk Knowledge, combining hazard and vulnerability, is the first pillar of an Early Warning System. Risk maps can represent an important tool for stakeholders from the transportation sector in a climate change, disaster risk reduction and sustainable

development agenda."

Page 1, Line 20: double citation for a same author

Second and third paragraphs in introduction section may be collapsed. The authors are invited to check similar occurrences on next paragraphs

Please, check the journal directives about citing/referencing – as written on Page 2/Line 4, "...as presented in (Yin & Xu (2010); Santos et al. (2019))", the included parenthesis looks inadequate for this kind of citation. The authors should consider this concern for the entire manuscript

Furthermore, paragraph of single phrase should be avoided (e.g., Page 2, Line 7 and more)

Last paragraph of Section 1 looks out of context

Page 2, Line 25: check decimal/thousands separator on English writing

Include a figure to express the study area location (South america ! Brazil ! State) may help the study/manuscript understanding

Page 3, Line 3: Since Herrmann et al. (2014) has more then one author, the pronoun "He" is inadequate

Page 3, Line10: Use acronym for institution citation, instead of Universidade Federal de Santa Catarina

The proportional symbol was wrongly chosen

Figure 2 caption is missing. What means the numbers 1 and 2 on Figure 2?

The conclusions should be improved/enhanced

The last paragraph of Section 3 looks out of context

R: In a new version, we are going to correct all these technical mistakes.

The authors may rise such problematic before state the use of Geo-graphs as a tool on flooding situations.

Any previous study or similar research (on the Geo-graph point of view) were cited. If this is a pioneer study, the authors should ensure and then highlight it

R: We thank the reviewer for this comment. In a new version, we are going to clarify it:

"According to Pregnolato et al. (2016), network models are typically aspatial, the emphasis has been on topological interactions rather than considering the geography of the hazard and infrastructure. Here, we use the concept and tools of a (geo)graph, a graph in a geographical space (Santos et al., 2017). Recently, this approach was applied for a mobility network analysis (Santos et al., 2019) and for a rainfall network analysis (Seron et al., 2019)"

Regarding a path dij, always is possible to find j from i? Are defined paths with null cost? If yes, how the efficiency is computed in such cases? Why the efficiency is inversely proportional to dij?

R: In this case, yes, it is a graph connected, there is always a path connecting i to j. And there is no edges nor paths with null cost.

A discussion about the vulnerability on k (the meaning behind the mathematical definition) should be included – what such model means? Page 3, Line 30: What means a vulnerability of 3%? How such value is interpreted/understood?

R: We thank the reviewer for this comment. In a new version, we are going to clarify it:

Method: "The topological vulnerability index associated to an element on a graph can be understood as the way the system reacts to a damage on this element. Although it is a measure associated to the element, assuming a possible value for each element, the vulnerability on complex networks brings information about the dynamics throughout the whole network (Santos et al., 2019)"

Results: "In Figure 1 there are several elements with low topological vulnerability index, but there are some elements with vulnerability approximately 3% and an element with vulnerability approximately 5%, so, a flood enabling the traffic on this highway's element can reduce the efficiency of this transportation network by approximately 5%. It is important to highlight that in this region there is a heavy flow of people and goods, with some important national and international ports and airports."

How Figure 1 was generated? The author may include a graph representation for "lengths" information used to achieve Figure 1;

R: Using the (geo)graph approach, we have represented the set of highways as a graph, calculated the properties and represented them back on the highways map. In Figure 1 is shown the topological vulnerability map for all highways in the study area.

Discussions regarding Figure 2 should be improved/enhanced

R: In a new version, we are going to clarify it, considering the figure itself and also a discussion about its importance:

"Figure 2 shows the vulnerability index map for a subset of the highways in the study area, and, also, the areas most susceptible to floods and a digital image of the terrain. In this subset, it is possible to see that there are some elements with high topological vulnerability index close to urban areas susceptible to flood. In this area, in the cities Rio Negrinho and Mafra, the BR-280 highway crosses the Negrinho River. This area is marked by several records of floods in the rainy season (hazard component), which makes traffic in the region unfeasible (impact) (UFSC, 2016)."

"This representation, considering both a vulnerability (as a topological index) and susceptible areas, can be an important tool for stakeholders from the transportation sector in a climate change, disaster risk reduction and sustainable development agenda. The Risk Knowledge, combining hazard and vulnerability, is the first pillar of an Early Warning System."

How the flood susceptible areas on Figure 2 were obtained?

R: In Rio Negrinho and Mafra the highway crosses the Negrinho River. The susceptible flood areas used in this study were mapped by the Geological Survey of Brazil (CPRM), based on a database of previous occurrences and in situ evaluation of physical characteristics.

How such areas affect the path's susceptibilities? Low susceptibilities are found near to flood areas – isn't expected the inverse behavior?

R: In this paper, as several other in Disaster Science literature, susceptibility is a property associated to a physical area, and, on the other hand, vulnerability is a set of properties associated with people or infrastructures in an area. In this paper, we combined them in order to produce a simple but useful risk map, able to inform where the most vulnerable elements are and if they are close to susceptible areas.

We do thank the reviewer for all the comments, the contributions were very important to improve this work.

Please also note the supplement to this comment:
https://www.nat-hazards-earth-syst-sci-discuss.net/nhess-2019-199/nhess-2019-199-AC1-supplement.pdf